# Dietary Intake of Pregnant Women with and without Inflammatory Bowel Disease in the United States

**DOI:** 10.3390/nu15112464

**Published:** 2023-05-25

**Authors:** Barbara C. Olendzki, Bi-Sek Hsiao, Kaitlyn Weinstein, Rosemary Chen, Christine Frisard, Camilla Madziar, Mellissa Picker, Connor Pauplis, Ana Maldonado-Contreras, Inga Peter

**Affiliations:** 1Department of Population and Quantitative Health Sciences, University of Massachusetts Chan Medical School, Worcester, MA 01655, USA; christine.frisard@umassmed.edu (C.F.); camilla.madziar@umassmed.edu (C.M.); 2Department of Nutrition, University of Massachusetts Amherst, Amherst, MA 01003, USA; bhsiao@umass.edu; 3Department of Genetics and Genomic Sciences, Icahn School of Medicine at Mount Sinai, New York, NY 10029, USA; kaitlyn.weinstein@mssm.edu (K.W.); rosemary.chen@mssm.edu (R.C.); mellissa.picker@mssm.edu (M.P.); inga.peter@mssm.edu (I.P.); 4Department of Medicine, University of Massachusetts Chan Medical School, Worcester, MA 01655, USA; connor.pauplis@umassmed.edu; 5Department of Microbiology and Physiology Systems, University of Massachusetts Chan Medical School, Worcester, MA 01655, USA; ana.maldonado@umassmed.edu

**Keywords:** diet, pregnancy, IBD, inflammatory bowel disease, dietary guidelines

## Abstract

Background: Pregnancy is a vulnerable time where the lives of mother and baby are affected by diet, especially high-risk pregnancies in women with inflammatory bowel disease (IBD). Limited research has examined diet during pregnancy with IBD. Aims: Describe and compare the diet quality of pregnant women with and without IBD, and examine associations between dietary intake and guidelines during pregnancy. Methods: Three 24 h recalls were utilized to assess the diets of pregnant women with IBD (*n* = 88) and without IBD (*n* = 82) during 27–29 weeks of gestation. A customized frequency questionnaire was also administered to measure pre- and probiotic foods. Results: Zinc intake (*p* = 0.02), animal protein (g) (*p* = 0.03), and ounce equivalents of whole grains (*p* = 0.03) were significantly higher in the healthy control (HC) group than the IBD group. Nutrients of concern with no significant differences between groups included iron (3% IBD and 2% HC met the goals), saturated fat (only 1% of both groups met the goals), choline (23% IBD and 21% HC met the goals), magnesium (38% IBD and 35% HC met the goals), calcium (48% IBD and 60% HC met the goals), and water intake (49% IBD and 48% HC met the goals). Conclusions: Most pregnant women in this cohort fell short of the dietary nutrients recommended in pregnancy, especially concerning for women with IBD.

## 1. Introduction

Pregnancy is a critical time for the intergenerational transmission of health [1,2,3,4]. Pregnant women with active inflammatory bowel disease (IBD), a chronic disease characterized by inflammation of the gastrointestinal tract [5] are considered to be at higher risk of poor pregnancy outcomes such as preterm birth, low birthweight or small for gestational age (SGA), spontaneous abortion, and stillbirth, and comprise an increased percentage of Cesarean deliveries compared to women in remission or without IBD [6,7,8,9]. The prevalence of IBD has been increasing worldwide [5]; thus, improving the health of pregnant women with IBD is essential to decreasing their risk for adverse pregnancy outcomes.

A balanced perinatal diet can support optimal health for pregnant women and have a long-term impact on their offspring [10,11,12,13]. Patients with IBD are already prone to nutrition deficiencies due to factors such as restrictive diets, nutrient loss, drug–nutrient interaction, and decreased absorption from the ileum [14]. Furthermore, reduced oral intake and chronic inflammation increases nutrient needs among IBD patients [15,16]. Two reports have explored the diets of pregnant women in the Norwegian Mother and Child Cohort (MoBa). The first study found that compared to pregnant women without IBD, pregnant women with IBD were less likely to adhere to a traditional Norwegian dietary pattern characterized by a high intake of lean fish or fish products, potatoes, rice porridge, cooked vegetables, and gravy, and were more likely to adhere to a Western dietary pattern with higher intake of foods and beverages rich in sugar and saturated fats [17,18]. Moreover, pregnant women with IBD who did adhere to the traditional Norwegian diet had lower odds of having an SGA infant [17]. The second study found that pregnant women with IBD consumed a lower proportion of protein from dairy products compared to pregnant women without IBD. In this case, a reduced intake of protein from dairy was associated with a lower risk of having an SGA infant [18].

Maternal diet during pregnancy has also been linked to the infant microbiome composition, which is critical for the priming of a balanced immune system during early life [19]. Importantly, our team has demonstrated that infants born to women with IBD have less diverse microbiomes and higher levels of fecal calprotectin (a biomarker of intestinal inflammation) compared to the infants of women without IBD [4,20]. Along with emerging reports demonstrating the mediating role of the gut microbiota in the effectiveness of dietary interventions for IBD management [21,22], this finding suggests that improving dietary patterns during pregnancy may beneficially modify the microbiome composition, thereby promoting both maternal and infant health. This hypothesis is being explored by the MELODY (Modulating Early Life Microbiome through Dietary Intervention in Pregnancy) Trial [12].

Diet has been increasingly integrated into IBD management, and studies demonstrate the effectiveness of dietary interventions for inducing IBD remission [23,24,25]. In adults, the specific carbohydrate diet (SCD); the Mediterranean diet; the low fermentable oligosaccharides, disaccharides, monosaccharides, and polyols (low FODMAP) diet; and the anti-inflammatory IBD (IBD-AID) diet are among those that have shown efficacy in reducing disease activity and symptoms [23]. Yet, informational resources on nutrition for pregnant women with IBD are sparse. The USDA MyPlate website focuses on a variety of food groups with only broad suggestions of foods and meal plans specific to pregnancy and postpartum needs [26]. The 2014 and 2017 American College of Obstetricians and Gynecologists (ACOG) guideline statements seem focused on nutrients that may be obtained by taking a prenatal vitamin, rather than on whole foods [27,28]. The 2019 American Gastroenterological Association’s Inflammatory Bowel Disease in Pregnancy Clinical Care Pathway report encourages nutrition consultation for specific nutrient deficiency and weight gain patterns in this population, but with few details on compliance to guidelines [29]. In keeping with these publications, pregnant women may hear only general advice from health care providers to take a prenatal vitamin, follow a healthy diet, limit caffeine intake, avoid alcohol and tobacco, and observe caution with seafood [30,31]. However, while a prenatal vitamin may be recommended in addition to a healthy diet, it cannot supply all the nutrients that are needed to promote healthy and low-risk pregnancies [32].

While diet can support IBD management, with the potential to positively benefit perinatal as well as longer-term health outcomes, little is known about the quality of dietary patterns among pregnant women with IBD in the United States (US), a country with a high prevalence of the disease. Therefore, the objectives of the current study are to describe the dietary patterns and diet quality of pregnant women with and without IBD living in the US, and to examine the associations between dietary patterns, diet quality, and dietary guidelines for pregnancy established by the Society for Obstetricians and Gynaecologists of Canada; the American College of Obstetricians and Gynecologists; the World Health Organization Guidelines; the Academy of Nutrition and Dietetics; the Royal College of Physicians of Ireland; the National Institutes of Health Daily Recommended Intake; and UpToDate [27,30].

## 2. Methods

We conducted a case–control study nested into our ongoing MELODY Trial, which is a prospective non-randomized diet intervention trial testing the effects of IBD anti-inflammatory diet (IBD-AID) during the third trimester of pregnancy on maternal IBD activity and microbiome composition in mothers and their babies [12]. Pregnant women with and without IBD were recruited nationwide for this trial. Study participants were identified by clinical research coordinators in outpatient gastrointestinal clinics; alternatively, pregnant women reached out if interested after seeing posts on the websites or Facebook accounts of the Crohn’s and Colitis Foundation or the Center for Applied Nutrition at the University of Massachusetts. Written informed consent was obtained from all eligible participants. The current case–control study examines dietary assessments conducted at the 27th–29th weeks of gestation prior to any dietary intervention, between January 2019 and December 2022.

The study was approved by the Institutional Review Boards at each institution (IRB docket #H00016462 at the University of Massachusetts Chan Medical School and #18–01206 at the Icahn School of Medicine). The inclusion criteria included: pregnant women carrying a singleton pregnancy, and a documented IBD diagnosis or lack thereof (for healthy controls, HC). The diagnosis of IBD was based on the patient’s history supported by clinical documentation. The exclusion criteria were an inability to provide informed consent, HIV/AIDS, multi-fetus pregnancy, fetal chromosomal or structural abnormalities, intrauterine growth restriction, active infection (including chorioamnionitis or sepsis), alcohol use disorder, renal disease, or a dietary regime that conflicts with the intervention diet. Additionally, pregnant IBD patients who had active perianal or extra-intestinal disease or were treated with antibiotic therapy or steroids at recruitment, as well as women scheduled for C-section prior to week 37, were excluded [12]. The final selection of participants is shown in Figure 1.

### 2.1. Dietary Assessment

We performed three 24 h dietary recalls (24 HRs) and a specially designed pre- and probiotic food frequency questionnaire for more detailed detection of food groups than provided by the 24 HR (IBD-AID FFQ) [33,34,35,36,37,38]. The 24 HR were performed using the University of Minnesota Nutrition Coordinating Center’s (NCC) Nutrition Data System for Research (NDSR) software (current version: NDSR, 2022, updated yearly) as previously described [12,39,40]. Specifically, trained dietitians administered 24 HR on two weekdays and one weekend, by phone, between 27 and 29 weeks of pregnancy. The 24 HR also included assessments of dietary supplements. The IBD-AID FFQ was self-administered online using REDCap, as previously reported by us [12]. The dietary assessments were conducted from 2019 to 2022.

### 2.2. Dietary Quality Assessment

Diet quality was estimated from the 24 HR recalls using the standard Alternative Healthy Eating Index—2010 (or AHEI-2010) score (range: 0–110), with higher scores representing healthier diets [39,41,42], and the IBD-AID FFQ score (range: 0–26) [12,37,43]. The IBD-AID FFQ was developed by Barbara Olendzki and her team at the Center for Applied Nutrition, Umass Chan Medical School, and addresses a gap in the nutrition information available from the 24 HR recalls, particularly with regard to pre-and probiotic foods. We found construct validity in using the IBD-AID FFQ, as pre- and post-dietary intervention changes correlated with bacterial abundance and serum cytokine levels [43]. The beneficial foods of the IBD-AID FFQ were matched with the food categories of the validated Alternate Healthy Eating index-2010 or AHEI-2010 [44]. Namely, the IBD-AID FFQ assesses the intake of 15 food groups and components. Beneficial Nutrient Score is calculated from all components and ranges from 0 to 26. Raw Score = [prebiotic foods] + [probiotic foods] + [Beneficial Nutrient Score] − [adverse foods]. The standard score eliminates the negative values, so if the raw score is <0, then the standard score is 0. If the raw score is >0, then the standard score is the raw score. In addition, the IBD-AID FFQ measures prebiotic foods (>3 servings/day), probiotic foods (>2 servings/day), and foods associated with gastrointestinal symptoms and poor IBD outcomes, including: refined carbohydrates (<2 servings per day), lactose (0 servings), certain grains (wheat, corn; 0 servings/day), processed foods (0 servings per day) and foods high in saturated (<7% of calories) or trans fats (0 servings/day).

We scored each beneficial food component (to correlate with the AHEI) from non-adherence = 0, to perfect adherence = 26. Pre- and probiotic foods were scored separately, with a perfect score being >3 and >2 servings per day, respectively. The IBD-AID FFQ total score = (prebiotic foods + probiotic foods + beneficial foods) minus adverse foods, with higher scores representing higher servings of beneficial foods minus adverse foods.

### 2.3. Statistical Methods

The demographic characteristics were presented using means and standard deviations for continuous variables and compared between pregnant women with and without IBD using a two-sample *t*-test. The categorical variables were described using counts and proportions, with *p*-values calculated via a Fisher’s exact test. To minimize the bias of a particular day where food intake is not typical, the reported servings were averaged across three 24 HR. The means and standard deviations summarized nutrients, components of interest from foods, and food group servings on the IBD-AID FFQ using two-sample t-tests quantifying between-group differences for normally distributed data and Wilcoxon rank sum tests for skewed outcome variables. The proportions and standard deviations described the proportion of participants who met the guidelines for nutrients at baseline, with chi-square tests measuring between-group differences.

## 3. Results

### 3.1. Participant Characteristics

The demographic characteristics of our study population, comprising 82 healthy controls (HC) and 88 participants with IBD (Crohn’s disease (CD) = 80, and ulcerative colitis (UC) = 8), are presented in Table 1. On average, women were 34 ± 4 years of age, predominantly white (91%) and non-Hispanic (90%). Most were married (93%), had a 4-year college degree or greater (90%), were employed full-time (71%), and had a household annual income of more than USD 100,000 per year (72%). Most were non-smokers (90%) and took a daily prenatal vitamin (91%). There were no significant differences between HC and IBD participants, except in profession and religious affiliation. The proportion of women with IBD who identified as Jewish was higher than the in the HC cohort (*p* < 0.001). The women with IBD reported working in more scientific technical professions than the HC group, while HC women reported working in more skill-, craft-, and health-based professions compared to IBD participants (*p* = 0.02). The average disease duration of the IBD participants was 14 years for CD and 10 years UC. We found 89% remission for our CD patients (three did not complete this section of the form so 4% were N/A), and 88% remission for our UC patients.

The list of IBD-directed medications is provided in Table 1.

### 3.2. Nutrient Intake and Dietary Quality for Pregnant Women with and without IBD

In total, we collected 496 24 HR at 27 and 29 weeks of pregnancy for the 170 women included in the study.

We estimated diet quality using the AHEI-2010 (scored 0–110), which incorporates components of evidence-based recommendations to identify future risk of chronic disease [40,44]. Overall, the participants in the study had a higher dietary quality (66.6 in IBD group, 67.9 in HC group) compared to the average American (47.6 ± 10.8) [44,45]. There were no differences in dietary quality between pregnant women with and without IBD.

Table 2 presents the average nutrients and components of interest from foods sources, excluding any dietary supplements. On average, the intake of nutrients was comparable between pregnant women with and without IBD, with some notable exceptions. The percentage of calories from monounsaturated fatty acids (MUFAs), and the intake of eicosapentaenoic acid (EPA) and docosahexaenoic acid (DHA), were significantly higher in pregnant women with IBD than HC. Conversely, healthy control participants reported a significantly higher intake of animal protein (their total protein intake was similar), whole grains, lactose, and zinc. There were no differences in calories consumed per group (approximately 2000 kcals/day).

The proportion of women meeting the pregnancy dietary guidelines for nutrients from food is shown in Table 3. The following dietary guideline goals were significantly lower in women with IBD than in the controls; 63% vs. 80% met the thiamine guideline (*p* = 0.01), and 38% vs. 56% met the B6 guideline (*p* = 0.02). Both the IBD and HC groups mostly met the guidelines for caffeine (94% and 96%, respectively) and taking a prenatal vitamin (93% and 90%, respectively). A total of 39% of women with IBD met the zinc guideline vs. 54% of the HCs (*p* = 0.05). Protein intake was not optimal for either group (with 66% meeting the guidelines). A total of 38% of women with IBD met the guidelines for total fat intake, and 45% met the guidelines in the HC group (*p* = 0.31). Nutrients of concern include iron (3% IBD and 2% HC met the goals), saturated fat (only 1% of both groups met the goals), choline (23% IBD and 21% HC met the goals), magnesium (38% IBD of 35% HC met the goals), calcium (48% IBD and 60% HC met the goals), and water intake (49% IBD and 48% HC met the goals), with no significant differences between the groups.

Table 4 presents the average food group servings of the IBD-AID FFQ for the subset of women who completed the questionnaire at baseline. A lower percentage of women from both groups completed this questionnaire, as it was self-administered and not facilitated by a dietitian (68/88, 77.3% IBD and 65/82, 79.3% HC). Prebiotics were significantly higher in mothers with IBD (6.3 vs. 4.7 in HC, *p* < 0.001), as were non-wheat fiber/grains (6.3 vs. 2.8 in HC, *p* < 0.001). Servings of adverse foods (i.e., higher in sugar, wheat, lactose, and saturated and trans fats) were lower in women with IBD than in the HCs (7.1 vs. 15.7, *p* < 0.001), and lean proteins (2.4 IBD vs. 3.4 HC, *p* = 0.04) and servings of beneficial beverages (apple cider, low-sugar beverages with added probiotics, juice (no added sugar), non-dairy milk, homemade smoothies, honey tea, tomato juice, V8 juice, coconut water, tea, and coffee substitutes (chicory root)) were significantly lower in pregnant participants with IBD than those without IBD (1.8 vs. 6.6, *p* < 0.001); however, total water intake was similar between groups. The Beneficial Nutrient Score (a calculated score of prebiotics, probiotics, overall dietary quality, and intake of foods thought to be adverse) was significantly lower in HC participants than in those with IBD (15.6 vs. 14.0, *p* = 0.04), as were the IBD-AID FFQ total raw and standard scores (16.4 IBD vs. 4.6 HC, *p* < 0.001; 16.9 IBD vs. 8.4 HC, *p* < 0.001, respectively).

## 4. Discussion

The current study addresses the gap in knowledge about the diets of pregnant women with IBD through an analysis of baseline data from The MELODY Trial. We observed that pregnant women with and without IBD do not consume most of the nutrients and food components recommended during pregnancy by established government and research-based organizations.

Specifically, we found that although most women reported taking a prenatal supplement, in hopes of supplementing the inadequate intake of nutrients from food sources, pregnant women IBD and the HC group fell short of most nutrients recommended in pregnancy from food sources alone. Of particular concern in women with IBD are the dietary micronutrients zinc, iron, calcium, magnesium, choline, folate, B6, B12, water, and fiber [46,47,48,49,50]. Patients with IBD require additional assistance to compensate for increased nutritional needs and poor absorption, whereby simply adding the nutrient does not guarantee that it will be well absorbed in the body [51,52].

Iron needs increase during pregnancy, especially in women with IBD, who may struggle with significant inflammation, anemia, and dysbiosis, leading to poor cardiovascular outcomes and suboptimal gestational weight gain [12,53,54,55,56,57]. The recommended daily allowance for pregnant women is around 27 mg per day. Many factors can influence iron absorption in the body, such as certain nutrient–nutrient interactions, including nutrient inhibitors (such as calcium) and enhancers (i.e., ascorbic acid) [58]. Furthermore, non-heme iron, primarily found in plant sources, is less easily absorbed by the body than heme iron, primarily found in animal sources, so the recommended amount of iron for vegetarians and vegans is 1.8 times greater [59]. In this study, only 3% of women with IBD and 2% of the HCs met the dietary guideline for (animal-based) iron. It is estimated that between 36 and 90% of people with IBD have iron-deficient anemia (IDA) [47] and that 15–20% of pregnant people have IDA [60]. This can lead to worse disease outcomes for both mothers and infants [61]. It is important that pregnant women meet the dietary guidelines set for iron first through food consumption, subsequently adding supplements as the need is determined.

The adequate intake of fiber is 28 g per day [62]. Fiber is typically not found in prenatal vitamins but is an especially important dietary component during pregnancy. Adequate fiber intake during pregnancy is crucial and may help alleviate iron-induced constipation. It is also helpful for reducing certain problems during pregnancy, such as inflammation, gestational diabetes mellitus, and cardiovascular outcomes [63,64]. However, even among the general population, dietary fiber intake falls below the recommended 28 g per day [56]. A study using 2001–2014 National Health and Nutrition Examination Survey (NHANES) data found that pregnant, non-lactating women aged 20–40 (*n* = 1003) had a mean total daily intake of 17.3 g of dietary fiber [65]. Only 33% of pregnant women with IBD in our study achieved the recommended fiber intake of >28 g per day. One mechanism by which diet can provide protection from IBD is through the addition of plant-based, fiber-rich foods that promote short-chained fatty acid-producing bacteria, which have been shown to support mucosal barrier integrity [23]. Adequate fiber intake important not only for women’s health during pregnancy [62,64,66,67,68,69], but also for preventing infant outcomes such as SGA, preterm birth, and fetal growth restriction [70].

Both groups were consuming an excess of foods with saturated fat, which can lead to an elevated risk of gestational diabetes [71]. Notably, the IBD group consumed less zinc and calcium than did the HC group. Low calcium and zinc intakes have been correlated with risk for poor outcomes for both mother and child [46,47,72]. Prenatal vitamins may not overcome a low dietary intake of these nutrients [73]. In our sample, predominantly white women with a college degree and who earned higher than the national average income consumes a fairly healthy diet before entering the study, which is why we do not see many significant between-group differences.

A recent study revealed that almost no supplements met nutritional needs in the doses that are required for pregnant women (without excess) [32]. Prenatal vitamins are recommended for pregnancy, especially to provide folic acid, EPA/DHA, iron, and vitamin D. Vitamins are, by definition, recommended to supplement the diet, not to replace the inclusion of the nutritious foods needed during pregnancy. Nutrients are digested and absorbed most effectively in the complex milieu of the foods themselves and the complementary enzymes and microbiota that facilitate absorption. Absorption is biologically complex, and simply adding a nutrient does not mean it will be well absorbed [51,52]. Therefore, many individuals with IBD require additional assistance to compensate for increased nutritional needs and poor absorption, as the nutritional needs of pregnancy for women with IBD may be uniquely challenging. Despite the excellent consensus recommendations by the International Organization for the Study of Inflammatory Bowel Diseases (IOIBD), there is no guidance for pregnancy with IBD or prenatal advice for the prevention of IBD. The IOIBD, based in large part on epidemiological studies, does not cover altering the textures of foods (such as pureeing fiber) for ease of absorption, one of many considerations that goes beyond the nutrients themselves and addresses malnutrition and malabsorption [74].

However, even with the limited existing evidence, healthcare providers appear to be inadequately counseling pregnant IBD patients on diet. Apart from referral to a dietitian for gestational diabetes, the prescription of dietary guidelines for pregnant women is lacking, increasing the risk for detrimental outcomes, especially for those with high-risk pregnancies. Current data show that only 37% of pregnant IBD patients reported receiving education from any physician about IBD in pregnancy [75]. Even worse, only 10% of patients reported having received pregnancy-specific information from their gastroenterologists, and of those who received information, 48% found the information to be insufficient [75]. Yet, several studies have demonstrated that those women who receive dietary counseling during pregnancy eat more fruits and vegetables, promoting the healthy growth and development of the fetus [76,77,78,79], suggesting that more dietary interventions are needed.

This apparent lack of patient education is not due to physician ignorance regarding pregnancy and IBD. In fact, when assessed with the Crohn’s and Colitis Pregnancy Knowledge Score (CCPKnow), 91.8% of physicians demonstrated very good knowledge, with gastroenterologists scoring the highest [75,80]. In contrast, only 10.3% of patients exhibited very good knowledge when assessed using the CCPKnow, with 44.8% demonstrating poor knowledge levels [80]. This discrepancy between physician and patient CCPKnow scores highlights the need for increased patient counseling, particularly from gastroenterologists, who exhibit the highest CCPKnow scores [80]. Healthcare providers should first evaluate pregnant individuals at risk of nutrient deficiency and excess, and subsequently provide evidence-based suggestions for supplementation [81]. Importantly, specific suggestions and menu plans with foods that contain essential nutrients and other components, such as fiber and pre-and probiotics, should be presented in actionable formats.

We acknowledge that the assessment of diet is prone to limitations, including self-report bias, under- or overestimation, memory bias, and weakness in the methodology. Further, we did not account for the influence of the environment, medication, dietary supplementation, or IBD activity status on nutrition, although most of our IBD patients were in remission. We conducted a large part of this study during the pandemic, when changes to food intake occurred, and this would have affected both arms of the study. This study is further limited to pregnant women in the United States of higher socioeconomic status, as they may have better access to medical care and foods that may not be generalizable to other groups and countries. However, our IBD and HC study groups were well-balanced regarding age, education, and income, suggesting that the reported differences (or lack thereof) in dietary intake are representative of this cohort.

## 5. Conclusions

While many women adhered to taking a prenatal supplement, both pregnant IBD participants and HC participants fell short of most dietary nutrients recommended in pregnancy through dietary sources alone, especially micronutrients and fiber. The consumption of animal protein, lactose, zinc, and whole grains was significantly lower in pregnant women with IBD compared to the HCs. Large epidemiological and dietary intervention studies are warranted to improve the nutritional recommendations for pregnant women with and without IBD while addressing malnutrition and malabsorption. Future research should consider pregnancy outcomes and the effects on offspring, and determine the causes of dietary deficiencies and excess, to ultimately inform and improve the quality of provider training and patient education, especially in the setting of IBD.

## Figures and Tables

**Figure 1 nutrients-15-02464-f001:**
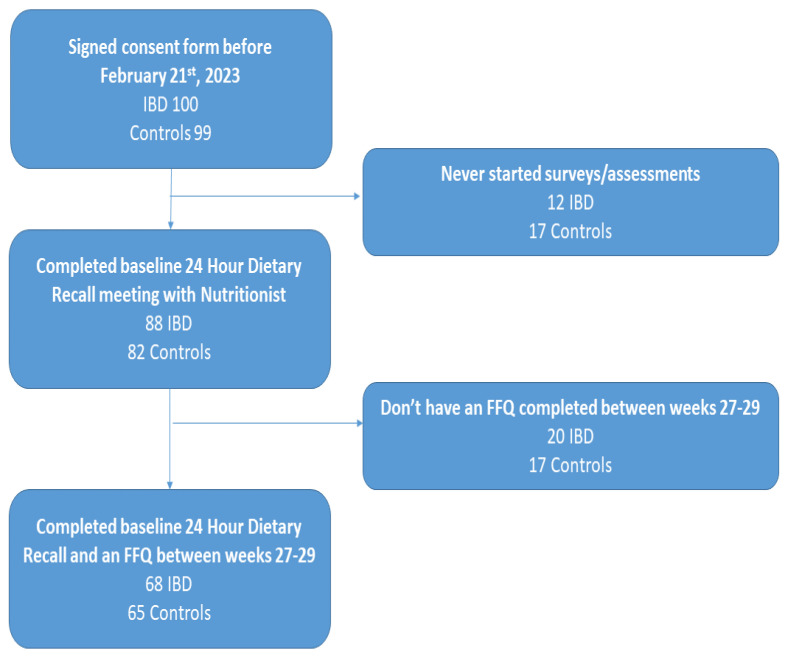
Participant Flow.

**Table 1 nutrients-15-02464-t001:** Demographic Characteristics of Healthy Controls vs. Pregnant Women with Inflammatory Bowel Disease at Baseline (*n* = 170).

	IBD *n* = 88	Healthy Controls *n* = 82	Total	*p*-Value
	*n* (%)	*n* (%)	*n* (%)	
Age—mean (SD)	33.7 (4.4)	34.4 (4.4)	34.0 (4.4)	0.32
Race				0.26
White	82 (94.3%)	70 (87.5%)	152 (91.0%)	
Black	2 (2.3%)	1 (1.3%)	3 (1.8%)	
Asian	1 (1.1%)	4 (5%)	5 (3.0%)	
Other	2 (2.3%)	5 (6.3%)	7 (4.2%)	
Hispanic or Latino descent				0.29
No	81 (93.1%)	70 (87.5%)	151 (90.4%)	
Yes	6 (6.9%)	10 (12.5%)	16 (9.6%)	
Jewish				<0.001 *
No	71 (88.8%)	49 (56.3%)	120 (71.9%)	
Yes	9 (11.3%)	38 (43.7%)	47 (28.1%)	
Marital status				0.85
Married	73 (91.3%)	82 (94.3%)	155 (92.8%)	
Single	3 (3.8%)	2 (2.3%)	5 (3.0%)	
Living with partner	3 (3.8%)	3 (3.4%)	6 (3.6%)	
Other	1 (1.3%)	0 (0.0%)	1 (0.6%)	
Education				0.38
High school graduate	0 (0.0%)	4 (4.6%)	4 (2.4%)	
Some college	5 (6.3%)	2 (2.3%)	7 (4.2%)	
Associate’s degree	3 (3.8%)	4 (4.6%)	7 (4.2%)	
Bachelor’s degree	24 (30.4%)	27 (31.0%)	51 (30.7%)	
Graduate or professional degree	46 (58.2%)	49 (56.3%)	95 (57.2%)	
Other	1 (1.3%)	1 (1.1%)	2 (1.2%)	
Work status				0.87
Employed full-time	55 (68.8%)	63 (72.4%)	118 (70.7%)	
Employed part-time	9 (11.3%)	10 (11.5%)	19 (11.4%)	
Homemaker (not looking for a job)	10 (12.5%)	6 (6.9%)	16 (9.6%)	
Disabled (unable to work)	1 (1.3%)	2 (2.3%)	3 (1.8%)	
Unemployed	3 (3.8%)	4 (4.6%)	7 (4.2%)	
Student	2 (2.5%)	2 (2.3%)	4 (2.4%)	
Type of work				0.02 *
Skill or craft	3 (7.5%)	6 (16.7%)	9 (11.8%)	
Scientific technical work	11 (27.5%)	1 (2.8%)	12 (15.8%)	
Service work	10 (25%)	10 (27.8%)	20 (26.3%)	
Health professional	16 (40%)	19 (52.8%)	35 (46.1%)	
Total annual household income				0.77
less than USD 20,000	2 (2.8%)	0 (0.0%)	2 (1.5%)	
USD 20,000–USD 39,000	2 (2.8%)	1 (1.7%)	3 (2.3%)	
USD 40,000–USD 59,000	4 (5.6%)	1 (1.7%)	5 (3.8%)	
USD 60,000–USD 79,000	5 (7.0%)	5 (8.5%)	10 (7.7%)	
USD 80,000–USD 99,000	9 (12.7%)	8 (13.6%)	17 (13.1%)	
USD 100,000 or more	49 (69.0%)	44 (74.6%)	93 (71.5%)	
Smoking status				0.45
Non-smoker	70 (87.5%)	79 (91.9%)	149 (89.8%)	
Ex-smoker	10 (12.5%)	7 (8.1%)	17 (10.2%)	
Intake of prenatal vitamins				0.58
No	8 (10.3%)	6 (7.1%)	14 (8.6%)	
Yes	70 (89.7%)	79 (92.9%)	149 (91.4%)	
IBD medication				
Aminosalicylates	21 (23.8%)	NA		
Anti-TNF	28 (31.8%)	NA		
Immunomodulators	4 (4.5%)	NA		
Oral corticosteroids	6 (6.8%)	NA		
Ustekinumab	16 (18.1%)	NA		
Vedolizumab	9 (10.2)	NA		

IBD—inflammatory bowel disease. * *p*-value < 0.05.

**Table 2 nutrients-15-02464-t002:** Nutrients and Components of Interest from Foods in Pregnant Women with Inflammatory Bowel Disease vs. Healthy Controls.

	IBD (*n* = 88)		Healthy Controls (*n* = 82)		
Nutrients	Mean	SD	Mean	SD	*p*-Value
Energy (kcal)	1994.6	461.1	2077.8	524.9	0.27
% Calories from Fat	37.8	6.8	36.3	6.9	0.16
% Calories from Carbohydrate	46.1	8.2	46.9	8.3	0.52
% Calories from Protein	16	3.5	16.56	4.6	0.36
% Calories from Alcohol	0.04	0.06	0.2	0.5	0.77
% Calories from SFA	12.5	3.3	12.8	3.2	0.57
% Calories from MUFA	14.0	3.6	12.8	3.1	0.03 *
% Calories from PUFA	8.0	2.2	7.5	2.0	0.14
Polyunsaturated to Saturated Fat Ratio	0.7	0.3	0.7	0.3	0.11
Animal Protein (g)	47.3	19.11	54.4	23.6	0.03 *
Vegetable Protein (g)	31.2	12.7	30.5	8.6	0.70
Total Dietary Fiber (g)	22.7	10.3	24.2	7.7	0.31
Soluble Dietary Fiber (g)	6.6	2.9	6.9	2.2	0.54
Insoluble Dietary Fiber (g)	16.0	8.0	17.2	6.1	0.27
Total Sugars (g)	97.3	37.8	102.0	42.7	0.44
Added Sugars (by Total Sugars) (g)	51.5	29.8	49.8	31.0	0.72
Glycemic Index (glucose reference)	58.4	4.6	57.9	4.1	0.45
Glycemic Load (glucose reference)	126.2	44.4	131.3	45.7	0.46
Total Grains (oz equivalents)	7.0	3.2	7.6	3.1	0.29
Whole Grains (oz equivalents)	1.6	1.3	2.1	1.6	0.03 *
Refined Grains (oz equivalents)	5.4	2.9	5.4	2.9	0.97
Lactose (g)	9.2	7.4	13.6	11.2	0.01
Sucrose (g)	44.5	21.4	45.1	23.7	0.87
Starch (g)	100.2	40.8	106.5	37.5	0.30
Total Folate (mcg)	431.0	153.7	460.4	143.0	0.20
Dietary Folate Equivalents (mcg)	537.0	217.2	583.5	218.6	0.17
Choline (mg)	359.7	145.5	354.0	128.5	0.79
Vitamin B-12 (cobalamin) (mcg)	4.1	2.3	4.3	2.0	0.66
Calcium (mg)	1033.9	354.3	1153.4	474.6	0.06
Magnesium (mg)	335.3	120.0	338.7	94.3	0.84
Iron (mg)	14.6	5.5	16.0	5.6	0.10
Zinc (mg)	10.5	3.8	11.8	3.8	0.02*
Copper (mg)	1.5	0.7	1.5	0.5	0.73
Selenium (mcg)	111.9	33.8	117.4	40.8	0.34
Sodium (mcg)	2984.9	715.2	3226.5	1112.9	0.09
Potassium (mg)	2568.4	915.6	2808.1	803.5	0.07
Omega-3 Fatty Acids (g)	1.78	1.23	1.8	0.87	0.73
PUFA 18:3 *n*-3 (alpha-linolenic acid [ALA]) (g)	1.8	1.78	1.8	0.87	0.39
PUFA 20:5 (eicosapentaenoic acid (EPA)) (g)	0.1	0.06	0.03	0.03	0.03 *
PUFA 22:6 (docosahexaenoic acid [DHA)) (g)	0.1	0.14	0.1	0.07	0.01 *
Water (g)	3126.9	987.1	3076.4	873.0	0.73

IBD—inflammatory bowel disease; SD—standard deviation; SFA—saturated fatty acids; MUFA—monounsaturated fatty acids; PUFA—polyunsaturated fatty acids. * *p*-value < 0.05.

**Table 3 nutrients-15-02464-t003:** Proportion of Women Meeting the Dietary Guidelines for Nutrients at Baseline (*n* = 170).

	IBD (*n* = 88)			Healthy Controls (*n* = 82)		
Guideline ^1^	Goal	(%)	SD	(%)	SD	*p*-Value *
Meets guideline for protein	71 g/day	66.0	0.5	66.0	0.5	0.99
Meets the guideline for total fat	20% to 35% calories	38.0	0.5	45.0	0.5	0.31
Meets the guideline for saturated fat	<7% of daily calories	1.0	0.1	1.0	0.1	0.96
Meets the guideline for EPA/DHA	1750 mg/3 days avg, or 583.33/day	15.0	0.4	5.0	0.2	0.03 *
Meets the guideline for carbohydrates	45 to 65% of caloric intake	58.0	0.5	65.0	0.5	0.37
Meets guideline for vitamin A	770 mcg/day from food	26.0	0.4	22.0	0.4	0.52
Meets the guideline for vitamin E	15 mg/day	26.0	0.4	20.0	0.4	0.31
Meets the guideline for vitamin C	85 mg/day	50.0	0.5	57.0	0.5	0.34
Meets the guideline for vitamin K	90 mcg/day	70.0	0.5	71.0	0.5	0.97
Meets the guideline for folate	600 mcg/day	13.0	0.3	15.0	0.4	0.68
Meets the guideline for iron	27 mg/day	3.0	0.2	2.0	0.2	0.71
Meets the guideline for calcium	1000 mg/day	48.0	0.5	60.0	0.5	0.12
Meets the guideline for choline	450 mg/day	23.0	0.4	21.0	0.4	0.75
Meets the guideline for caffeine	<200 mg/day	94.0	0.2	96.0	0.2	0.53
Meets the guideline for thiamine	1.4 mg/day	63.0	0.5	80.0	0.4	0.01 *
Meets the guideline for niacin	18 mg/day	70.0	0.5	79.0	0.4	0.19
Meets the recommendation for B6	1.9 mg/day	38.0	0.5	56.0	0.5	0.02 *
Meets the recommendation for B12	2.6 mcg/day	72.0	0.5	83.0	0.4	0.08
Meets the recommendation for zinc	11 mg/day	39.0	0.5	54.0	0.5	0.05 *
Meets the guideline for magnesium	360 mg/day	38.0	0.5	35.0	0.5	0.77
Meets the guideline for copper	1000 mcg/day	84.0	0.4	89.0	0.3	0.35
Meets the guideline for total fiber	>28 g/day	33.0	0.5	44.0	0.5	0.14
Meets the recommendation for water	3000 g/day, or 101.44 oz	49.0	0.5	48.0	0.5	0.87
Meets the recommendation of taking a prenatal vitamin (*n* = 85 IBD, *n* = 78 controls)	Yes	93.0	0.3	90.0	0.3	0.47
Alternative Healthy Eating Index/HEI Items		Mean	SD	Mean	SD	
AHEI-10 score (0 = 110)	0–110	67.9	12.3	66.5	12.0	0.45
Total fruit servings in cup equivalents	3	1.1	1.2	1.1	1.0	0.88
Total vegetable servings in cup equivalents	5	1.8	1.2	2	1.0	0.26

IBD—inflammatory bowel disease; SD—standard deviation. * *p*-value < 0.05. ^1^ Guidelines were selected from a review of the following organizations: Society for Obstetricians and Gynaecologists of Canada; American College of Obstetricians and Gynecologists; World Health Organization Guidelines; Academy of Nutrition and Dietetics; Royal College of Physicians of Ireland; National Institutes of Health Daily Recommended Intake; and UpToDate.

**Table 4 nutrients-15-02464-t004:** Average Daily Food Group Servings of the IBD-AID Food Query.

		IBD (*n* = 68)		Healthy Controls (*n* = 65)		
Variable	Number of Servings for Optimal Score	Mean Servings	SD	Mean	SD	*p*-Value
Prebiotic score ^1^	≥3	6.3	5.4	4.7	10.2	<0.0001 *
Probiotics score ^2^	≥2	1.6	1.8	1.6	3.4	0.80
Adverse foods score ^3^	0	7.1	4.5	15.7	38.0	<0.0001 *
Vegetable score	5	3.6	3.2	4.5	13.0	0.6
Fruit score	3	2.1	1.9	2.2	2.2	0.92
Nuts, seeds, and oils score	2	2.0	2.2	1.8	2.6	0.89
Lean protein score ^4^	4	2.4	2.0	3.4	12.5	0.04 *
Fiber/grains score ^5^	3	6.3	6.3	2.8	5.2	<0.0001 *
Probiotic dairy score ^2^	3	1.1	1.2	1.2	1.4	0.27
Non-caloric fluids score		6.7	3.5	7.2	3.6	0.45
Beneficial beverages score ^6^	6	1.8	3.7	6.6	6.8	<0.0001 *
Condiments score		0.3	0.4	0.7	2.9	0.63
Alcohol score		0	0	0.4	3.0	0.47
Foods with unknown effects ^7^		0.3	0.5	1.1	4.8	0.04
IBD FFQ Beneficial Nutrient Score ^8^	26	15.6	5.0	14.0	4.6	0.04
IBD AID total raw score ^9^		16.4	12.9	4.6	24.3	<0.0001 *
IBD AID total standard score ^10^		16.9	12.3	8.4	7.1	<0.0001 *

IBD-AID—inflammatory bowel disease anti-inflammatory diet. * *p*-value < 0.05. ^1^ Prebiotics are foods containing fiber that feed commensal organisms. ^2^ Probiotics are fermented foods that contain live bacteria. ^3^ Adverse foods include ultra-processed foods and foods high in added sugars. N optimal adverse foods goal is zero servings per day and counts negatively toward total score. ^4^ Lean protein score includes beans/legumes, seafood, and poultry. ^5^ Fiber/grains include foods such as oats, barley, and miso. ^6^ Beneficial beverages include beverages such as those with added probiotics, non-dairy milks, homemade smoothies, no-sugar-added fruit and vegetable juices, coconut water, tea sweetened with honey, etc. ^7^ Foods with unknown effects have yet to be determined in research. ^8^ Beneficial Nutrient Score is calculated from all components and ranges from 0 to 26. ^9^ Raw Score = [prebiotic] + [probiotics] + [Beneficial Nutrient Score] − [adverse]. ^10^ The standard score eliminates the negative values, so if the raw score is <0, then the standard score is 0. If the raw score is >0, then the standard score is the raw score.

## Data Availability

The data presented in this study are available on request from the corresponding author. The data are not publicly available due to data privacy restrictions.

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
