# Peer review of "Dietary Intake of Pregnant Women with and without Inflammatory Bowel Disease in the United States"

_nutrients, 2023, doi:10.3390/nu15112464_

Round 1
Reviewer 1 Report
· Title: Please include the country where the study was performed.
· The abstract does not reflect the most important results of the paper. P-values or statements about the significance of results should be reported. The place where the study was conducted, the date of sampling took place, the age of participants and the type of study design and statistical analysis should be clearly stated. The conclusion does not propose a clear direction for future studies.
· Line 64-78: Can authors please expand on the effect of diet on the gut microbiota, and whether it has implications for IBD in both mothers and children? I would recommend referring to these articles (Nutrients. 2022 Oct 3;14(19):4113; Proc Nutr Soc. 2021 Sep 23;1-15).
· Line 87-92: The study design should be clear defined. Is it case-control study? More details on the sample recruitment are also required.
· Line 93-103: Please include a clear diagram showing the final selection of participants.
· Authors should include hypothesis about the non-significant differences in the discussion section.
· Line 309-310: The limitations of the study should be clearly described.
· The conclusion is too short. The implications of results and the directions for future research are not clearly discussed.
· Please update the references list. Some references were very old (Ref # 1, 8, 33, 35, 62). Please also follow the journal guideline for referencing.
Author Response
We thank the reviewers for their insightful comments and suggestions. We thoroughly revised the manuscript to improve clarity, provide detailed explanations, and updated references. Below are our responses to each of the reviewer's concerns/ critiques/ suggestions: (see attached)
- Title: Please include the country where the study was performed.
: The title has been changed to include the country where the study took place.
- The abstract does not reflect the most important results of the paper. P-values or statements about the significance of results should be reported. The place where the study was conducted, the date of sampling took place, the age of participants and the type of study design and statistical analysis should be clearly stated. The conclusion does not propose a clear direction for future studies.
: We apologize for the lack of details in the abstract. We have now updated the abstract to include statements of significance, the mean age of participants, the place where the study was conducted, the sampling dates, the study design, and future directions.
- Line 64-78: Can authors please expand on the effect of diet on the gut microbiota, and whether it has implications for IBD in both mothers and children? I would recommend referring to these articles (Nutrients. 2022 Oct 3;14(19):4113; Proc Nutr Soc. 2021 Sep 23;1-15).
: These two references have been added. Additional implications for the impact of dietary factors on the microbiome of pregnant mothers and children have been added (See lines 63-67).
“Along with emerging reports demonstrating the mediating role of the gut microbiota in the effectiveness of dietary interventions on IBD management [21,22], this finding suggests that improving dietary patterns during pregnancy may beneficially modify the microbiome composition, thereby promoting both maternal and infant health.
- Line 87-92: The study design should be clearly defined. Is it a case-control study? More details on the sample recruitment are also required.
: We apologize for the lack of clarity. We have now included the study design in the abstract (lines 30-31, “Methods: We conducted a case-control study from 2019-2022.”) and in the methods section (lines 103-104). …” We conducted a case-control study nested into our ongoing MELODY Trial, which is a prospective non-randomized diet intervention trial…”
- Line 93-103: Please include a clear diagram showing the final selection of participants.
: We appreciate the suggestion. We have now included a flowchart to show the recruitment and final subject selection for the study (See Figure 1) and refer to lines 126.
“Final selection of participants is shown in figure 1.”
- Authors should include hypotheses about the non-significant differences in the discussion section.
: We have now added a discussion/hypothesis about the non-significant differences in the discussion section. See lines 313-316.
“In our sample, predominantly white women with a college degree and higher than the national average income in were all eating a fairly healthy diet before entering the study which is why we do not see many significant between-group differences.”
- Line 309-310: The limitations of the study should be clearly described.
: we agree and now have a clear description of the limitations in lines 357-367.
“We acknowledge that the assessment of diet is prone to limitations, including self-report bias, under or overestimation, memory bias, and weakness in methodology. Further, we did not account for environmental, medication, dietary supplementation, or IBD activity status influences on nutrition, although most of our IBD patients were in remission. We conducted a large part of this study during the pandemic, when changes to food intake occurred, though this would have affected both arms of the study. This study is further limited to pregnant women in the United States of higher socioeconomic status, as they may have better access to medical care and foods that may not be generalizable to other groups and countries. However, our IBD and HC study groups were well-balanced regarding age, education, and income suggesting that the reported differences (or lack thereof) in the dietary intake are representative of this cohort. “
- The conclusion is too short. The implications of results and the directions for future research are not clearly discussed.
: Thank you for pointing this out. We have now included a paragraph summarizing the implications of the results and the future directions. See lines 369-378.
“While many women adhered to taking a prenatal supplement, both pregnant IBD participants and HC participants fell short of most dietary nutrients recommended in pregnancy through dietary sources alone, especially micronutrients and fiber. Consumption of animal protein, lactose, zinc, and whole grains was significantly lower in pregnant women with IBD compared to HC. Large epidemiological and dietary intervention studies are warranted to improve the nutritional recommendations for pregnant women with and without IBD while addressing malnutrition and malabsorption. Future research should consider pregnancy outcomes and effect on offspring and determine causes of dietary deficiencies and excess, ultimately to inform and improve the quality of provider training and patient education, especially in the setting of IBD.”
- Please update the references list. Some references were very old (Ref # 1, 8, 33, 35, 62).
: We agreed with the reviewer about citing old references in the manuscript. We thoroughly revised the references and for every old (original) reference, we have cited we also cited newer references that support the original findings. We decided to honor the original research by citing it.
When reference 1 (from 1990) is used, it is supported by references 2, 3, and 4 (from 2016, 2011, and 2020).
When reference 8 (from 1998) is used, it is supported by references 6, 7, and 9 (from 2010, 2007, and 2021).
When reference 36 and 38 (previously 33 and 35) (from 1978 and 1999) are used, they are also supported by references, 33, 34, and 37 (from 2017, 2009, and 2001).
When reference 67 (previously reference 62) (from 1997) is used, it is also supported by references 62, 64, 66, 68, and 69 (from 2018, 2007, 2007, 2015, and 2021).
- Please also follow the journal guideline for referencing.
: We apologize for this oversight. We have updated the format of the references to comply with the journal guidelines.

Reviewer 2 Report
This is a very interesting topic and a lot of recent publications have reviewed the role of diet in IBD, but very few disussed dietary intake in pregnant woman.
Could you specify who developped the dietary questionnaire and how it was validated?
Was this retrospective or prospective?
During what time period was this study done?
What was the response rate for the questionnaire?
What was disease severity?
What medications were they on?
What was the disease duration?
Please add to the discussion the IOIBD recommendations concerning dietary intake of specific food groups in IBD patients.
Thanks.
Author Response
We thank the reviewers for their insightful comments and suggestions. We thoroughly revised the manuscript to improve clarity, provide detailed explanations, and updated references. Below are our responses to each of the reviewer's concerns/ critiques/ suggestions: (see attached)
REVIEWER 2:
This is a very interesting topic and a lot of recent publications have reviewed the role of diet in IBD, but very few discussed dietary intake in pregnant women.
- Could you specify who developed the dietary questionnaire and how it was validated?
: We have now provided more details on the dietary questionnaire in lines 198-230,
“The IBD-AID FFQ was developed by Barbara Olendzki and her team at the Center for Applied Nutrition, Umass Chan Medical School and addresses a gap in nutrition information available from the 24hr recalls, particularly with regards to pre-and probiotic foods. We found construct validity in using the IBD-AID FFQ as pre- and post-dietary intervention changes correlated with the bacterial abundance and serum cytokine levels [43]. The IBD-AID FFQ beneficial foods are matched with the food categories of the validated Alternate Healthy Eating index-2010 or AHEI-2010 [44]. Namely, the IBD-AID FFQ assesses the intake of 15 food groups and components that are listed in Table 4. Footnotes to Table 4 describe the scoring. In addition, the IBD-AID FFQ measures prebiotic foods (>3 servings/day), probiotic foods (>2 servings/day), and foods associated with gastrointestinal symptoms and poor IBD outcomes including: refined carbohydrates (<2 servings per day) , lactose (0 servings), certain grains (wheat, corn= 0 servings/day), processed foods (0 servings per day) and foods high in saturated (<7% of calories) or trans fats (0 servings/day).
We scored each beneficial food component (to correlate with the AHEI) from non-adherence = 0, to perfect adherence = 26. Pre- and probiotic foods were scored separately, with a perfect score being >3 and >2 servings per day respectively. The IBD-AID FFQ total score = (prebiotic foods + probiotic foods + beneficial foods) minus adverse foods, with higher scores representing higher servings of beneficial foods minus adverse foods.
and in footnotes to table 4.
“1Prebiotics are foods containing fiber that feed commensal organisms.
2Probiotics are fermented foods that contain live bacteria.
3 Adverse foods include ultra-processed foods and foods high in added sugars. An optimal adverse foods goal is zero servings per day and counts negatively toward total score.
4Lean protein score includes beans/legumes, seafood, poultry.
5Fiber/grains include foods such as oats, barley, miso.
6Beneficial beverages include beverages such as those with added probiotics, non-dairy milks, homemade smoothies, no sugar added fruit and vegetable juices, coconut water, tea sweetened with honey, etc.
7Foods with unknown effect: have yet to be determined in research.
8Beneficial Nutrient Score is calculated from all components, as 0-26. .
9Raw Score = [prebiotic]+[probiotics]+[beneficial nutrient score]-[adverse].
10The standard score eliminates the negative values, so if the raw score is < 0, then the standard score is 0. If the raw score is > 0, then the standard score is the raw score.”
- Was this retrospective or prospective?
: We apologize for the lack of clarity. The parent study is an ongoing prospective dietary intervention trial. We have updated the methods section accordingly (See line 153).
“We conducted a case-control study nested into our ongoing MELODY Trial, which is a prospective non-randomized diet intervention trial...”
- During what time period was this study done?
: The participants for this study were recruited between January 2019 and December 2022. The parent MELODY Trial is ongoing. Please see abstract, “We conducted a case-control study from 2019-2022. and methods, lines 168-169.
“The current case-control study examines the dietary assessments at the 27-29 weeks of gestation prior to any dietary intervention, between January 2019 and December 2022.”
- What was the response rate for the questionnaire?
: We have now added the questionnaire response rate in Table 4, lines 345-348.
“A lower percentage of women from both groups completed this questionnaire, as it was self-administered and not facilitated by a dietitian (68/88, 77.3% IBD and 65/82, 79.3% HC).”
- What was disease severity?
: We have included the disease severity to assess disease activity (lines 270-271).
“We found 89% remission for our CD patients (3 did not complete this section of the form so 4% are not available), and 88% remission for our UC patients. “
- What medications were they on?
: We have now included information about medication use at the bottom of Table 1.
|
Drug Class |
Sum of Number of Participants |
|
Aminosalicylates |
21 |
|
Anti-TNF |
28 |
|
Immunomodulatory Meds |
4 |
|
Oral Corticosteroids |
6 |
|
Ustekinumab |
16 |
|
Vedolizumab |
9 |
|
Grand Total |
84 |
- What was the disease duration?
We have now included information about disease duration in lines 274-276.
“The average disease duration of the IBD participants was 14 years for CD and 10 years UC.“
- Add to the discussion the IOIBD recommendations concerning the dietary intake of specific food groups in IBD patients.
: We appreciate the suggestion. We are now including a discussion of the IOIBD recommendations in lines 483-491.
“Despite the excellent consensus recommendations by the International Organization for the Study of Inflammatory Bowel Diseases (IOIBD), there is no guidance for pregnancy in IBD or prenatal advice for prevention of IBD. The IOIBD, based in large part on epidemiological studies, does not cover textures of foods (such as pureeing fiber) for ease of absorption, one of many considerations that goes beyond the nutrients themselves and addresses mal-nutrition and malabsorption [74].”

Round 2
Reviewer 1 Report
No further comments.
N/A
Author Response
Thank you for this suggestion. The p-values for differences between groups has been added to the abstract.
